# Association between meteorological factors and the prevalence dynamics of Japanese encephalitis

**Taotian Tu**[1ꙮ], **Keqiang Xu**[2ꙮ], **Lei Xu**[3,4], **Yuan Gao**[3], **Ying Zhou**[1], **Yaming He**[1], **Yang Liu**[1], **Qiyong Liu**[3], **Hengqing Ji**[1]*, **Wenge Tang**[1]*

**1** Chongqing Municipal Center for Disease Control and Prevention, Chongqing, China, **2** College of Computer and Information Engineering, Henan Normal University, Xinxiang, Henan Province, China, **3** State Key Laboratory of Infectious Disease Prevention and Control, National Institute for Communicable Disease Control and Prevention, Chinese Center for Disease Control and Prevention, Beijing, China, **4** Department of Earth System Science, Ministry of Education Key Laboratory for Earth System Modeling, Tsinghua University, Beijing, China

ꙮ These authors contributed equally to this work.
* twg@cqcdc.org (WT); hengqingji@163.com (HJ)

**Data Availability Statement:** The data of this study can be found at https://doi.org/10.17026/dans-xm3-7qdm.

**Funding:** This research was supported by the State Key Laboratory for Infectious Disease Prevention

## Abstract

Japanese encephalitis (JE) is an acute infectious disease caused by the Japanese encephalitis virus (JEV) and is transmitted by mosquitoes. Meteorological conditions are known to play a pivotal role in the spread of JEV. In this study, a zero-inflated generalised additive model and a long short-term memory model were used to assess the relationship between the meteorological factors and population density of *Culex tritaeniorhynchus* as well as the incidence of JE and to predict the prevalence dynamics of JE, respectively. The incidence of JE in the previous month, the mean air temperature and the average of relative humidity had positive effects on the outbreak risk and intensity. Meanwhile, the density of all mosquito species in livestock sheds (DMSL) only affected the outbreak risk. Moreover, the region-specific prediction model of JE was developed in Chongqing by used the Long Short-Term Memory Neural Network. Our study contributes to a better understanding of the JE dynamics and helps the local government establish precise prevention and control measures.

## Introduction

Japanese encephalitis (JE) is an acute zoonotic disease caused by the Japanese encephalitis virus (JEV, include five serotypes) and is transmitted by mosquitoes, primarily *Culex tritaeniorhynchus*. It is estimated that three billion persons live in countries where the JE virus is endemic worldwide. At present, JE is prevalent mostly in 24 countries of Southeast Asia and the Western Pacific; about 68,000 clinical cases are reported annually, and the case fatality rate is 25 to 30%. About 30 to 50% of JE survivors have permanent neurological sequelae, imposing a heavy burden on public health and society [1–3]. Historically, JE was serious disease in China. The number of JE cases has declined significantly Since the implementation of the nationwide immunization program in the 1970s [4]. To date, JE is considered one of the class

and Control Independent Fund (Contract no. 2018SKLID304). This research was partially supported by the donations from Delos Living LLC and the Cyrus Tang Foundation of Tsinghua University.

**Competing interests:** The authors have declared that no competing interests exist.

B notifiable infectious diseases [5]. In recent years, adults have been the population primarily affected [6]. However, JE still remains an important public health issue in China, with approximately a half of the reported cases worldwide [1].

Chongqing, China, is one of the areas with a high incidence of JE as its natural conditions, such as climate and environment, are suitable for the reproduction of mosquitoes. The incidence of JE fluctuated between 1.22 and 3.66 per 100,000 population from 1996 to 2006 in Chongqing. The vaccine for JE has been included in the Expanded Program on Immunization in Chongqing since 2007, which indicates that this vaccine has been included in the National Immunization Program for the routine vaccination of school-age children. Since then, the incidence of JE in Chongqing has decreased from 1.09 per 100,000 population in 2008 to 0.09 per 100,000 population in 2017. Although it decreased by 91.7% from 2007 to 2018, the infection rate is still the highest in Chongqing compare to other cities in China [7].

Recently, Daniel et al. found that rainfall (one-month lag), minimum temperature (six-month lag) and the Southern Oscillation Index (six-month lag) are positively associated with JE [8]. The spatial and temporal trends of the incidence of JE in the city of Chongqing are associated with temperature and rainfall [9]. Lin et al. found that the occurrence of JE is significantly associated with increasing temperature and relative humidity in Taiwan on the basis of Poisson's regression analysis and a case-crossover study [10]. Zhang et al. showed that meteorological variables are significantly associated with the incidence of JE between 2012 and 2014 using the Bayesian conditional auto-regressive model [11]. Prediction and early warning based on multiple factors have become an interesting topic in relation to the prevention and control of mosquito-borne infectious diseases, such as malaria [12]. Previous studies have carried out to develop JE prediction models, including the compartment model and time series model. Rubel et al. showed that the compartment model presented is able to quantitatively describe the process of JE dynamics [13]. The time series model used the relationship in the sequential lag time series to predict the incidence of JE, such as Methods Autoregressive integrated moving average (ARIMA) models [14]. However, the compartment model was used for some deterministic problems and required idealized hypothetical conditions [15]. Therefore, the existing compartment model has certain defects in predicting the trend of infectious disease. In numerous cases, the time series model did not consider the relationship between infectious disease.

This study addressed the limitation by a) analyzing the correlation between meteorology, mosquito density and the incidence of JE, and b) combining JE surveillance data, meteorological data and mosquito density data with the Long-Short Term Memory (LSTM) model to predict the JE incidence accurately. The results of this study can provide scientific information that can be used by the local government to establish precise prevention and control measures.

## Materials and methods

### Study area

Chongqing is located in the southwest of China (Fig 1A) and in the upper reaches of the Yangtze River. It has a sub-tropical monsoon humid climate, with an annual average temperature between 16˚C and 18˚C and an average annual rainfall of 1,000–1,350 mm. The eastern and southern parts of Chongqing rely on two large mountains (Daba and Wuling Mountains), and the northwest and central areas consist of mainly hills and low mountains. Numerous rivers are found in the region, and the mainstream of the Yangtze River runs through the whole territory from the west to the east. Because of the diverse ecological environment and rich vegetation in Chongqing, it is conducive for the reproduction of mosquitoes.

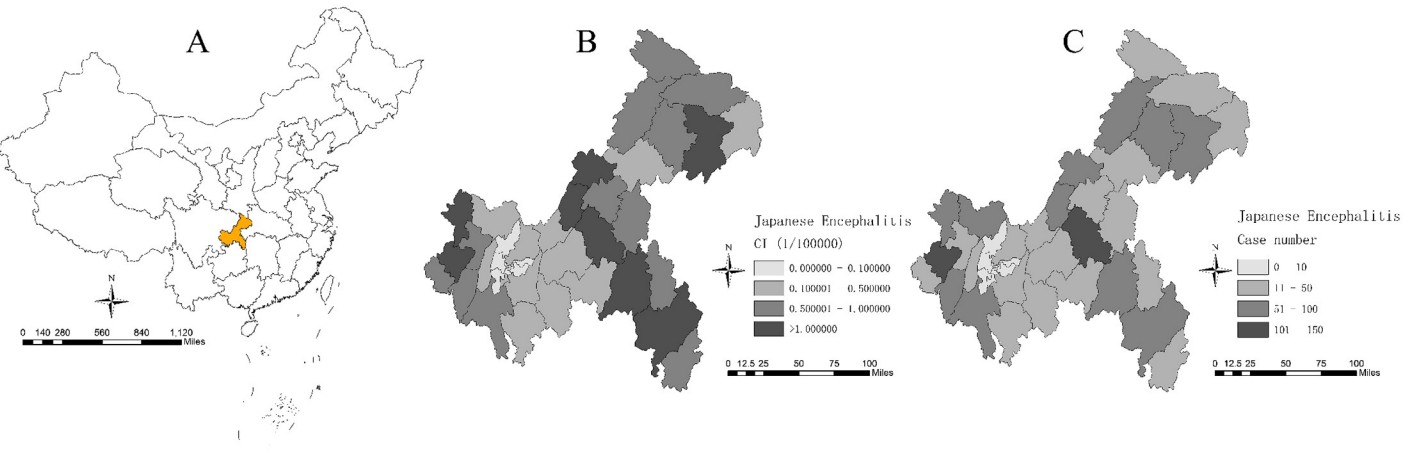

**Fig 1. Spatial distribution of the study area and Japanese encephalitis (JE) cases.** (A) The location is in Chongqing, China. (B) Spatial distribution of JE cases at the county level from 2007 to 2019 in Chongqing. (C) Cumulative incidence (CI) of JE cases at the county level from 2007 to 2019 in Chongqing.

## Data collection

Human JE incidence data in Chongqing from 2007 to 2019 were available from the National Notifiable Disease Surveillance System (NNDSS). Data on mosquitoes were obtained from Chongqing Center for Disease Control and Prevention (CQCDC), including *Culex tritaeniorhynchus* density in human houses (CDH), *Culex tritaeniorhynchus* density in livestock sheds (CDL), density of all mosquito species in human houses (DMSH) and density of all mosquito species in livestock sheds (DMSL). Meteorological data were obtained from National Meteorological Information Center (NMIC) (http://data.cma.cn/site/index.html), including mean air temperature, maximum air temperature, minimum air temperature and average of relative humidity. In total, 10 valid attributes were considered in our study, and the feature parameters are shown in Table 1. We summarized the data into a monthly scale. Human JE incidence data were aggregated per month and matched to monthly surveillance data of adult mosquito population density and monthly meteorological data for the model training and prediction. All the data required by this study is available at https://doi.org/10.17026/dans-xm3-7qdm.

**Table 1. The feature parameters used in this study.**

| Parameters | Symbol |
|---|---|
| Human JE cases per month | $D$ |
| Mean air temperature | $T_{mean}$ |
| Maximum air temperature | $T_{max}$ |
| Minimum air temperature | $T_{min}$ |
| Average of relative humidity | $H_{mean}$ |
| Precipitation | $P$ |
| the *Culex tritaeniorhynchus* density in human houses | CDH |
| the *Culex tritaeniorhynchus* density in livestock sheds | CDL |
| the density of all mosquito species in human houses | DMSH |
| the density of all mosquito species in livestock sheds | DMSL |

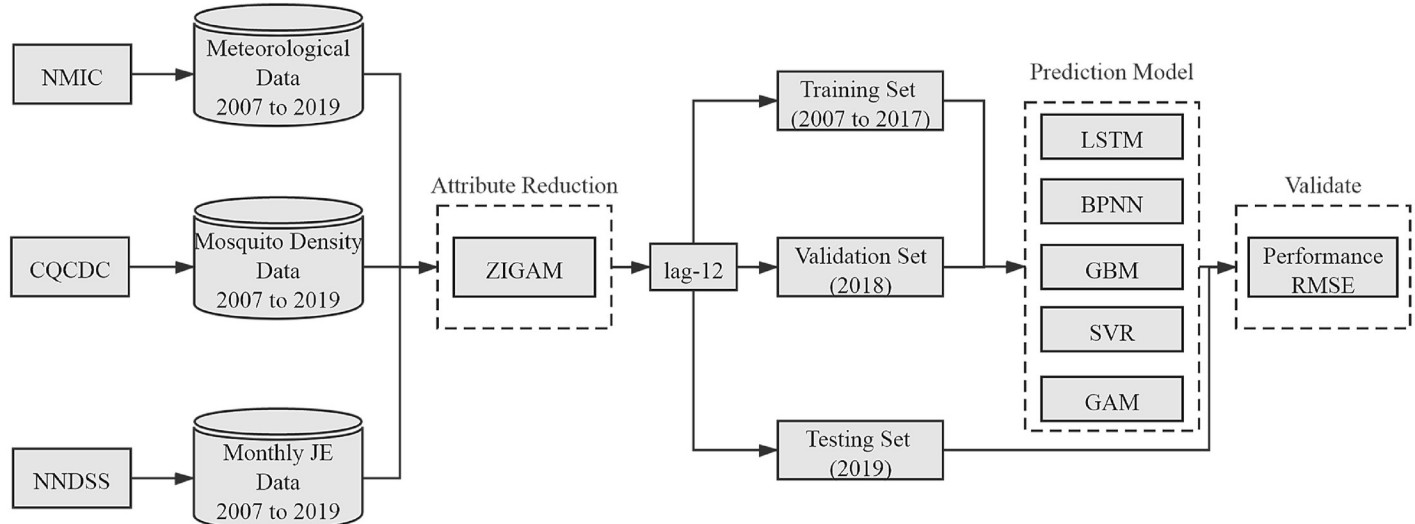

**Fig 2. Summarized workflow for the construction of the LSTM-based forecasting model for JE cases and its comparison with other candidate models.** NMIC: National Meteorological Information Center; CQCDC: Chongqing Center for Disease Control and Prevention; NNDSS: National Notifiable Disease Surveillance System; BPNN: Back Propagation Neural Network; GBM: Gradient Boosting Machine; SVR: Support Vector Regression; GAM: Generalized Additive Model.

## Statistical modelling

The overall framework of this study is illustrated in Fig 2. This study analyzed the correlation between meteorology, mosquito density and the incidence of JE by using the Zero-Inflated Generalized Additive Model (ZIGAM) and constructed four candidate prediction models of JE, including LSTM model, Back Propagation Neural Network (BPNN), Gradient Boosting Machine (GBM), Generalized Additive Model (GAM) and Support Vector Regression (SVR). Based on the inverse logarithm of model outputs, the model performance and prediction accuracy were measured by RMSE.

ZIGAM is widely used in applied statistics, particularly for modelling non-linear effects of covariates in scientific and quantitative studies. Some scholars successfully demonstrated the effects of climatic conditions on the mosquito density and dengue transmission rates by used ZIGAM [16]. Data on JE were zero-inflated (Fig 1). Therefore, the incidence dynamics of JE were analyzed with ZIGAM, which consisted of a binomial and a lognormal part, which was used to explore the relationships between the incidence of JE as well as mosquito density and meteorological factors using the COZIGAM package (version 2.0.4) of R (version 3.5.1) [17]. The maximum degree of freedom for the smooth terms was set to 4 in the model selection. We used an approximation of the log of the marginal likelihood (*logE*) as a model selection criterion, which was correlated to the Bayesian information criterion. Models with higher *logE* were preferred.

LSTM is a special kind of Recurrent Neural Network (RNN). The RNN algorithm are well suited for multivariate time series data, has the ability to learn patterns and extract features from data. LSTM has been widely used in a disease diagnosis, language identification and marine temperature due to its high accuracy [18–20]. Moreover, the model can be used to predict the incidence of infectious diseases [21]. In this study, the model was used to predict the incidence of JE. When truncating the gradient where it does not cause harm, the LSTM can learn to bridge minimal time lags in excess of 1,000 discrete-time steps by enforcing constant error flow through constant error carousels within special units [22]. In order to improve the practicability of the model, we use the number of monthly cases and local environmental variables that are 12 months behind as dependent variables. The model we built can predict the

trend of JE in the coming year. According to the ZIGAM study on the correlation between environmental variables and the incidence of JE, we chose the mean air temperature, average of relative humidity and DMSL as the input variables of the candidate models. The monthly data from 2007 to 2017 was used as the training set, the monthly data of 2018 was used as the verification set, and the monthly data of 2019 was used as the test set. In order to improve the prediction effect, make the model output multiple predictions by using the Dropout that can calculate the uncertainty of the LSTM forecasts and then calculate the prediction intervals.

The SVR model has shown an excellent performance for prediction data of time series. Specifically, we implemented an $\varepsilon$-SVR approach and tried values ranged from 0.05 to 1.0 with a span of 0.05 for the parameter *C*. Finally, we set the *C* parameter to 0.5 corresponding to the lowest RMSE value. For the BPNN model, an optimal parameter (set the number of layers to 3, the number of neurons to 6 and the learning rate to 0.02) was selected to avoid overfitting and improve the predictive performance. For the GAM model and the GBM model, the parameters of training used the default values in the python package.

Most of the experiments were run in the hardware environment with 64-bit Windows, a 3.0 GHz, Intel Core i5-8500 CPU. The BPNN model in this study were modeled through sknn library of python (version 0.7). The GBM model and SVR model in this study were modeled through scikit-learn library of python (version 0.22.2). The LSTM model in this study were modeled through TensorFlow (version 1.13.1), which is Google's released application programming interface for deep learning.

## Results

### Outcomes of ZIGAM

A total of 1,531 JE cases were reported from 2007 to 2019 in Chongqing (Fig 1B). The descriptive analysis results are shown in S1 Table. The cumulative incidence (CI) increased from 0 to 2.21 per 100,000 population (Fig 1C). The monthly time series plots of all variables are shown in Fig 3. A seasonal pattern was observed among all variables.

Data from 2007 to 2019 were used to explore the correlation between the incidence of JE and other variables. We used the variables with no significant correlation (correlation coefficient < 0.6) in the model (S2 Table). The final ZIGAM of the JE dynamics showed that the incidence of JE in the previous month had positive effects on the outbreak risk (i.e. on the probability of incidence being > 0; Fig 4A) and outbreak intensity [i.e. on ln(incidence), given that the incidence is > 0; Fig 4E]. The mean air temperature (> 16˚C) and average of relative humidity had positive effects on both outbreak risk and intensity (Fig 4B, 4C, 4F and 4G). Meanwhile, the DMSL sheds has a significant, approximately linear, and positive association with outbreak risk (Fig 4D).

### Outcomes of the LSTM model

The LSTM model can produce a prediction interval by using the dropout method in the prediction process. This method makes the model more practical. The root-mean-square error (RMSE) was utilised to evaluate the effect of prediction. The prediction results showed that the LSTM neural network models were indeed able to predict the high peaks in JE cases for these years compared to other prediction models (Fig 5 and S3 Table). In addition, Using the prediction models, we predicted the number of JE cases from January to December in 2020 (Fig 5).

According to the model performance for the JE prediction periods of Chongqing, the LSTM model had the smallest RMSE values, which mean the discrepancy between observed value and the value predicted under the LSTM model was smallest (Fig 5 and S3 Table). The results suggested that the LSTM model outperformed other compared models and was chosen as the optimal model in this study.

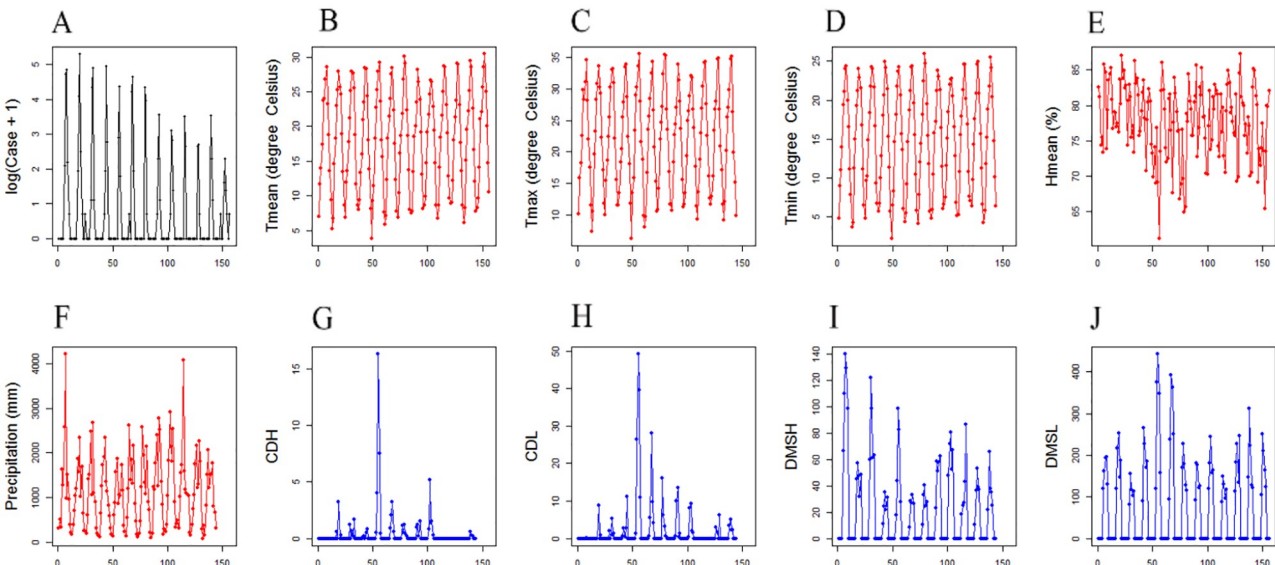

**Fig 3. Monthly time series plots in Chongqing from 2007 to 2019.** Monthly time series of the JE incidence (A), the mean air temperature (B), the maximum air temperature (C), the minimum air temperature (D), the average of relative humidity (E), the precipitation (F), the *C. tritaeniorhynchus* density in human houses (CDH) (G), the *C. tritaeniorhynchus* density in livestock sheds (CDL) (H), the density of all mosquito species in human houses (DMSH) (I) and the density of all mosquito species in livestock sheds (DMSL) (J) respectively.

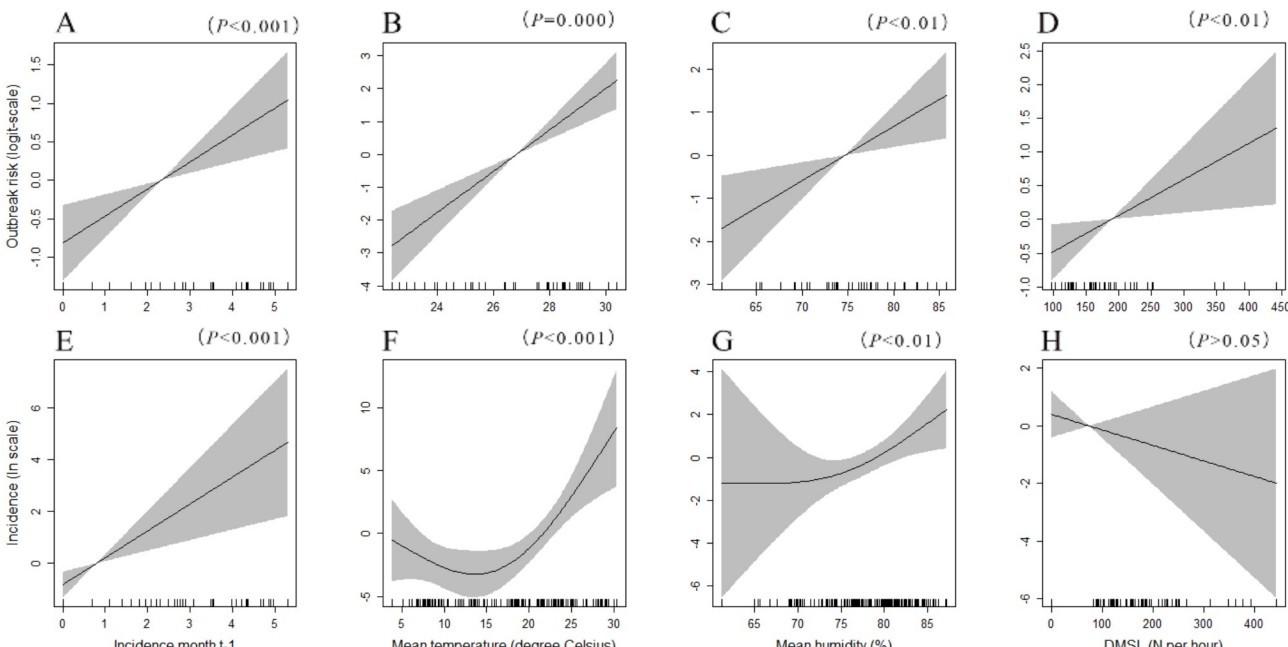

**Fig 4. Analysis of potentially non-linear influences on the incidence of JE in Chongqing based on data from 2007–2019.** In order to explain the zero inflation, (A) – (D) depict a separate binomial sub-model that quantifies predictor effects on the outbreak risk (logit-scale probability of incidence > 0) and (E) – (H) depict a lognormal sub-model that quantifies predictor effects on the outbreak intensity when an outbreak occurs [ln(incidence)].

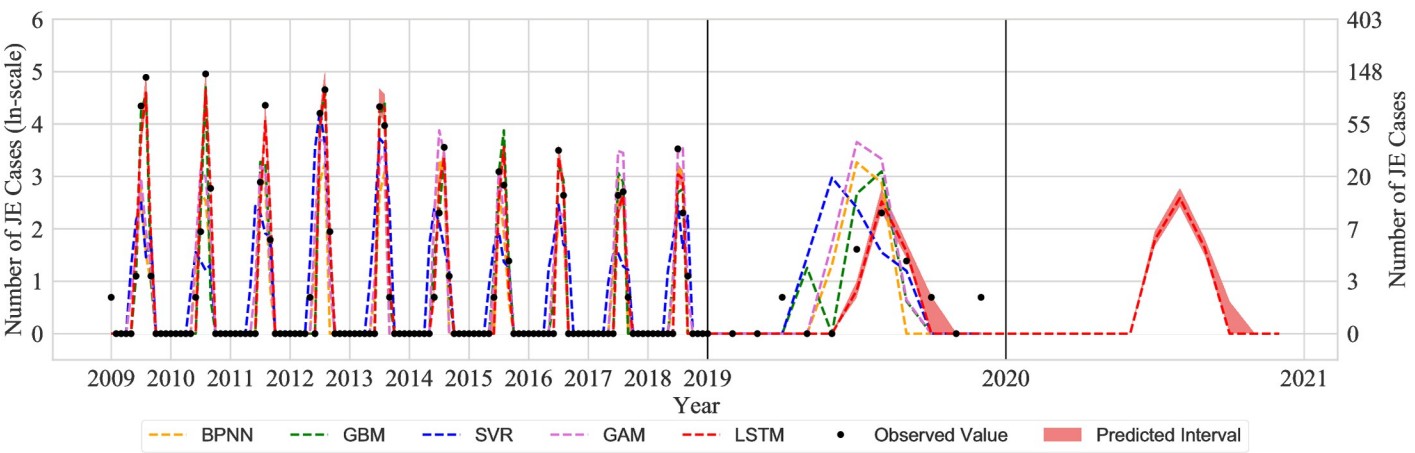

**Fig 5. Time series of the incidence of JE in Chongqing.** Data from 2008–2017 were used to train the LSTM model, data from 2018 were used as validation set and data from 2019 were utilised for model testing. In the figure, the observed values are represented by the black point, the predicted values of BPNN are represented by yellow line, the predicted values of GBM are represented by green line, the predicted values of SVR are represented by blue line, the predicted values of GAM are represented by purple line, the predicted values of LSTM are represented by red lines and the predicted interval of LSTM is represented by a light red area.

## Discussion

Chongqing, China, has a high incidence of JE, and a total of 1,531 cases were reported from 2007 to 2019. In this study, we not only analysed the relationship between meteorological factors and the incidence of JE using ZIGAM but also predicted the number of JE cases in Chongqing using the LSTM model. We found that the LSTM model was effective in predicting the incidence of JE.

Some studies found links between JE transmission and climate variations [23–26]. It is important to study the impact of weather on the transmission of Japanese encephalitis, as global climate change may directly or indirectly influence the mosquito density and the virus, as well as people's behaviours [27]. In our study, the previous-month mean air temperature and previous-month average of relative humidity were found to have positive effects on the outbreak risk and intensity. The previous-month DMSL only affected the outbreak risk. This result is agreement with other studies in general [14]. Our study suggest that meteorological variables are mixed in their effect on the transmission of JE. One or two meteorological factors may play greater role in JE case occurrence or transmission than others and this finding coincides with other published study [9, 28].

*Culex tritaeniorhynchus* is the main vector of the JE virus in the study area, and its life cycle varies depending on the climatic. Monthly mean air temperature was found to be associated with the elevated JE incidence in Chongqing, China. The result of our study showed that the number of JE cases began to increase at temperatures was greater than 16˚C. Similarly, Lin et al. found that the number of JE cases started to increase at temperatures of 22˚C in Taiwan [10]. Higher temperatures, within limits, lead to more rapid development of larvae, shorter times between bloodmeals, and faster incubation times for viral infections within mosquitoes. As a result, increases in temperature allow mosquito populations to reach higher levels faster, and to be maintained for longer, thereby increasing the opportunities for viral transmission [29]. For example, for Japanese encephalitis, only 14% of mosquitoes were infected when temperatures were 18–22˚C, but the figure reached 80% when temperatures were 26–30˚C [30], which further verified our findings.

In our results, the average of relative humidity also had a positive impact on the transmission of JE. Relative humidity influences longevity, mating, dispersal, feeding behaviour and oviposition of mosquitoes [31]. At higher humidity, mosquitoes generally survive for longer and disperse further. Therefore, they have a greater chance of feeding on an infected animal and surviving to transmit a virus to humans or other animals. *Culex tritaeniorhynchus* target large animals for blood extraction, including cattle and swine, and its trajectories are mostly distributed in the wild and livestock sheds. Swine is an important reservoir host and amplification host of JEV and are the main source of JE infection. Our findings indicated that the DMSL only affected the outbreak risk in Chongqing. The positive association with DMSL was consistent with findings of other studies [32, 33].

Climatic factors can change the living and proliferation environment of mosquitoes and lead to temporal and spatial changes in mosquito density. Meteorological factors such as temperature and humidity can affect the whole life cycle of mosquitoes, accelerate or delay the growth and development of mosquito larvae, and then affect the abundance of adult mosquitoes, and further affect the abundance of next-generation mosquitoes [34, 35]. The density of the mosquito also means the probability of biting the host carrying the Japanese encephalitis virus and transmitting the virus to humans [36]. In terms of epidemiology, the total length of the development of mosquito larvae, the external incubation period of Japanese encephalitis virus in adult mosquitoes and the internal incubation period in humans is about one month [23, 37], which is consistent with the experimental results of our model.

"Deep Learning", a branch of artificial intelligence, allows autonomously learning how viruses spread using raw observation data. Compared with the traditional statistical models, the deep learning method have many advantages, the most prominent of which is that deep learning models can automatically learn the information contained in the data without manually setting parameters such as thresholds value [22, 38]. The neural network model is established by setting the structural parameters. Adjusting the structural parameters of the model can obtain more accurate prediction results. If the established model is applied to the same task in other regions, it is only necessary to retrain the model by using the data of the other region without adjusting any parameters of model. In addition, the LSTM recursive neural network can use memory and gate units to truncate the gradient without damaging it and can bridge a large number of discrete-time steps to achieve the time series rule in a longer period of learning. Therefore, many different models have been developed by using LSTM, such as speech recognition model, forecasting model of air quality, forecasting models of infectious disease [21, 22, 38–40].

The ability to explore the relationship between climate and the incidence of infectious disease using ZIGAM has been demonstrated by previous research [17], and the LSTM neural network model has outstanding predictive power of infectious diseases compared to DNN and ARIMA [41]. In this study, we used the ZIGAM model to explore the relationship between the factors associated with JE; based on the factors with a relatively high correlation coefficient, such as mean air temperatures and mosquito density. Using the LSTM models, we assessed the number of JE cases from January 2019 to December 2020. The prediction of JE indicated that the high-risk season is in July and August. The results were consistent with that of the incidence of JE in Chongqing. Therefore, we emphasise that a set of models must be utilised to predict the number of cases in the subsequent year to improve the precision of prediction. Thus, attention should be paid and immediate actions must be taken during these months, and interventions such as health promotion and education, surveillance and control of mosquitoes and vaccination must be implemented immediately.

However, our research also had some limitations. First, the LSTM model takes a large amount of time for training compared; but the impact is not significant since the data collected

in this study were from a small-sized dataset. Second, we did not consider local potential related factors such as vegetation coverage and population movements, requiring further study and improvement in the future work.

## Conclusions

This study extended quantifying relationship between JE and meteorological variables based on the latest statistical analysis of multiyear time series and built a reliable prediction model of JE in Chongqing, China. The incidence of JE in the previous month, the mean air temperature and the average of relative humidity had positive effects on the outbreak risk and intensity. Meanwhile, the density of all mosquito species in livestock sheds (DMSL) only affected the outbreak risk. Moreover, we built a prediction model of JE by using LSTM in Chongqing, which enabled us to accurately predict monthly JE incidence using JE surveillance data and environmental data, including meteorology and mosquito density. According to the predicted data of model, the government can track epidemic dynamics to carry out targeted prevention and control measures. We conjecture that the result of this study could be applied more extensively in further researches.

## Supporting information

**S1 Table. Descriptive analysis of variables from 2007 to 2019 in Chongqing.**
(DOCX)

**S2 Table. Correlation analysis using Pearson's correlation test.**
(DOCX)

**S3 Table. Comparison of model performance using the Root Mean Square Error (RMSE).**
(DOCX)

## Acknowledgments

We would like to thank the staff at the Center for Disease Control and Prevention of Wanzhou District and Fengdu County for their support in collecting data during the study.

## Author Contributions

**Data curation:** Taotian Tu, Keqiang Xu, Ying Zhou, Yaming He, Yang Liu.

**Funding acquisition:** Wenge Tang.

**Methodology:** Yuan Gao.

**Resources:** Hengqing Ji.

**Software:** Keqiang Xu, Lei Xu.

**Supervision:** Taotian Tu, Qiyong Liu.

**Writing – original draft:** Taotian Tu.

**Writing – review & editing:** Lei Xu, Hengqing Ji, Wenge Tang.

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
