## [Decision Letter · Decision Letter 0]

24 Aug 2020

PONE-D-20-17377

Association between climate factors and the prevalence dynamics of Japanese Encephalitis

PLOS ONE

Dear Dr. Tu,

Thank you for submitting your manuscript to PLOS ONE. After careful consideration, we feel that it has merit but does not fully meet PLOS ONE’s publication criteria as it currently stands. Therefore, we invite you to submit a revised version of the manuscript that addresses the points raised during the review process.

We look forward to receiving your revised manuscript.

Kind regards,

Ram K. Raghavan

Academic Editor

PLOS ONE

Journal Requirements:

2. We note that Figure 1 in your submission contain map images which may be copyrighted. All PLOS content is published under the Creative Commons Attribution License (CC BY 4.0), which means that the manuscript, images, and Supporting Information files will be freely available online, and any third party is permitted to access, download, copy, distribute, and use these materials in any way, even commercially, with proper attribution. For these reasons, we cannot publish previously copyrighted maps or satellite images created using proprietary data, such as Google software (Google Maps, Street View, and Earth). For more information, see our copyright guidelines: http://journals.plos.org/plosone/s/licenses-and-copyright.

2.1. You may seek permission from the original copyright holder of Figure 1 to publish the content specifically under the CC BY 4.0 license. 

2.2. If you are unable to obtain permission from the original copyright holder to publish these figures under the CC BY 4.0 license or if the copyright holder’s requirements are incompatible with the CC BY 4.0 license, please either i) remove the figure or ii) supply a replacement figure that complies with the CC BY 4.0 license. Please check copyright information on all replacement figures and update the figure caption with source information. If applicable, please specify in the figure caption text when a figure is similar but not identical to the original image and is therefore for illustrative purposes only.

"This research was supported by the State Key Laboratory for Infectious Disease

Prevention and Control Independent Fund (Contract no. 2018SKLID304). This

research was partially supported by the donations from Delos Living LLC and the

Cyrus Tang Foundation of Tsinghua University."

"YES - Specify the role(s) played."

Additionally, because some of your funding information pertains to commercial funding, we ask you to provide an updated Competing Interests statement, declaring all sources of commercial funding.

In your Competing Interests statement, please confirm that your commercial funding does not alter your adherence to PLOS ONE Editorial policies and criteria by including the following statement: "This does not alter our adherence to PLOS ONE policies on sharing data and materials.” as detailed online in our guide for authors  http://journals.plos.org/plosone/s/competing-interests.  If this statement is not true and your adherence to PLOS policies on sharing data and materials is altered, please explain how.

Please include the updated Competing Interests Statement and Funding Statement in your cover letter. We will change the online submission form on your behalf.

Reviewers' comments:

Reviewer's Responses to Questions

**Comments to the Author**

1. Is the manuscript technically sound, and do the data support the conclusions?

Reviewer #1: Partly

Reviewer #2: Partly

2. Has the statistical analysis been performed appropriately and rigorously? 

Reviewer #1: No

Reviewer #2: I Don't Know

3. Have the authors made all data underlying the findings in their manuscript fully available?

Reviewer #1: Yes

Reviewer #2: Yes

4. Is the manuscript presented in an intelligible fashion and written in standard English?

Reviewer #1: No

Reviewer #2: Yes

5. Review Comments to the Author

Reviewer #1: The manuscript titled Association between climate factors and the prevalence dynamics of Japanese Encephalitis incidences (in terms of mosquito density) by Tu et al. presents an approach of nonlinear statistical modeling in predicting the JE dynamics over the region Chongqing (southwest part in China). Author considers the data for the period of 13 years for JE incidences and meteorological par ammeters.

However, the article is written very simply, and I observe not up to the standard of the journal. There are some grammatical issues along lacking the proper definition of study gap.

1. The term climate factor should be replaced with ‘meteorological factor’.

2. The author applies GAM to study the relationship for vector density, and long short term memory model (LSTM) for JE incidence with met. Parameters. No justification for adopting two different modeling aspect.

3. The predictive ability of the model should be written abstract line, and author avoid to do this.

4. The Introduction section is not structured. Globally published literature is missing. The hierarchy should be, global aspect, then continental, then country, then province/district. The author fails to perform this hierarchy. Some

5. Authors wrote, “However, no studies on the prediction of the incidence of JE, which is based on 84 factors such as mosquito density and climate, have been carried out” (pg. 4, line 83-84). In what context the author states this sentence, global or local. Around the globe, researchers have already contributed the stochastic modeling skill for association of JE cases and meteorological parameters (Lin et al., 2012; Pisudde et al., 2017). In what respect the author would like to state that. Justify and clarify.

6. Line #83-88. Is there only the meteorological factors that affect JE vector density for JE incidences? The several other factors like land use land cover changes, built dam, water bodies, socioeconomic status of the exposed population, intervention schemes/policy. Justify, why authors do not take into account, all these?

7. Why author opt for stochastic approach over the deterministic approach, the compartment modeling (deterministic) is also another way to model perfectly. Justification needed.

8. Line #89, instead of Methods, there should be Study Area, Data and Methods. The further subsection may be there.

9. The temporal resolution of data is missing in the ‘Data Collection’ section. For the ease of the scientific community, this should be there. A table may be presented here, for data sources of JE incidences as well as meteorological variables, along with the temporal resolution (daily/weekly/fortnightly/monthly/yearly).

10. The models’ outcome in the result section is not expressed much. The author was only direct to see the tables and figures. The crucial observations are not expressed.

11. The result and methods are very inconsistent. The result outcomes are self-contradictory. The result need to restructure.

12. Discussion is poorly written. Author try to make this very lengthy instead of technical discussion and suggestion. The scientific approach is missing in the discussion part.

13. The conclusion section is missing.

Lin, H., Yang, L., Liu, Q., Wang, T., Hossain, S.R., Ho, S.C., Tian, L., 2012. Time series analysis of Japanese encephalitis and weather in Linyi City, China. Int. J. Public Health 57, 289–296. https://doi.org/10.1007/s00038-011-0236-x

Pisudde, P.M., Kumar, P., Sarthi, P.P., Deshmukh, P.R., 2017. Climatic Determinants of Japanese Encephalitis in Bihar State of India : A Time-Series Poisson Regression Analysis. J. Commun. Dis. 49, 13–18. https://doi.org/10.24321/0019.5138.201729

Reviewer #2: The manuscript presents the results of a correlative analysis between selected climate variables and case incidence of Japanese Encephalitis in central China. The paper addresses a topic that is appropriate for the journal and potentially supplies important information for both basic scientific understanding of environmental influences on disease and for potential application to controlling the disease in this area. The paper compares several methods for deriving correlative models between climate and JE incidence, and does in fact provide the reader with an evaluation of the relative effectiveness of these models. However, the paper paper is frustratingly vague and difficult to follow at times, particularly in its coverage of the basic methodology and in its summary of how the climate variables are related to disease incidence.

The major issue with the paper is that it’s just very difficult to follow exactly what the authors did. Figure 2 shows a flow chart of the methods, showing the sources of data and the general flow of the analysis. It would be helpful to include a table in the body of the paper (rather than the supporting materials) that shows exactly w hat meteorological data are included. There are hints in the paper (temperature extrema seem to be one of the variables) but the reader never really knows for sure, and this becomes a problem later in trying to suss out what the findings of the study really are. All of the data are apparently funneled through the ZIGAM method, but the methods section of the paper never really explains what this is or what it does. It is defined on lines 79-80, and some references are cited. Additionally, in lines 130-141 some additional information is provided. Unfortunately, as a person wholly unfamiliar with this technique, this information was of little use. This may not be the case for readers more familiar with this type of analysis, but for those who are not knowledgeable in this area of analysis, the entire methodology description is not very helpful, at best. Since the entire analysis funnels though this technique, I would strongly urge the authors to write a more complete explanation of what this method does, and why it was chosen for this analysis. Otherwise, the reader is left with the frustrating task of searching through several more papers in order to understand what is in this one.

I had similar issues with the four methods for developing predictive models, LSTM, BPNN, GBM, and SVR. The way the paper is written seems to suggest the LSTM is somehow considered a standard in this type of analysis, and that the other three methods are presented as potentially superior alternatives? Is this correct? IF so, I would suggest that this be made clearer to the reader. On page 7, the paragraph beginning on line 142, some descriptive material for LSTM is provided, but again, in such a way as to be of little help to an unfamiliar reader. I would also note that there is a lot of detail about the other three methods (BPNN, GBM, and SVR) that is simply glossed over. I have only limited experience with neural networks and gradient boosting, but have worked rather extensively with support vector regression, enough to understand that parameter tuning and kernal basis functions are crucial to understanding the effectiveness of them. Absolutely no indication of any of this information is provided. At minimum, one should report the kernal used and the gamma and r parameters. Ideally, the method for determining these parameters should be provided. I suspect that similar information should be provided about the other two methods (GBM, BPNN). Without this type of information, the analysis cannot be replicated, nor does omitting them help other researcher who may wish to use this approach for similar analyses.

In addition to the lack of relevant detail in the methods section, the actual results also lack some specificity. The purpose of the paper was to model the relationship between climate and JE incidence, and to predict the occurrence of cases. Judging from Figure 5, the latter goal was achieved. Unfortunately, it is not really clear how it was achieved. Which climate variables were most relevant, and how relevant were they? Hard to really tell from the body of the manuscript. In the abstract (lines 27-30) it is noted that mean temperature and humidity are important (as is density of mosquito species in livestock sheds, a variable whose provenance is not very clear). In the body of the manuscript, though, this information is very difficult to separate out. Clearly, the authors have developed a method that is potentially useful for predicting JE in this study area, and potentially useful for researchers who may want to understand this disease (and possibly other, similar, mosquito-borne diseases) in other places. Without much clearer elucidation of their methods, this would be difficult to do. One of my “acid tests” for reading a manuscript is, “given similar data, could I repeat their methods, even if I had never used them before.” In this case the answer is no. Even if I were to track down all of the literature cited to support the methods, It’s still not clear how the authors reached their conclusions.

6. PLOS authors have the option to publish the peer review history of their article (what does this mean?). If published, this will include your full peer review and any attached files.

Reviewer #1: **Yes: **Dr. Praveen Kumar

Reviewer #2: No

---

## [Author Response · Author response to Decision Letter 0]

3 Nov 2020

November 3, 2020

PLoS ONE

Dear Editors,

On behalf of my co-authors, we thank you very much for giving us an opportunity to revise our manuscript, we appreciate the editor for their meticulous working attitude.

We redrew Figure 1 using the ArcGIS (version 10.6). Therefore, the figure has no copyright issues.

I look forward to hearing from you again.

Best regards,

Taotian Tu (taotiantu@sina.cn)

Chongqing Municipal Center for Disease Control and Prevention, Chongqing, China;

Wenge Tang (twg@cqcdc.org)

Chongqing Municipal Center for Disease Control and Prevention, Chongqing, China.

Response to Reviewer #1:

We would like to thank the reviewer for careful and thorough reading of this manuscript and for the thoughtful comments and constructive suggestions, which help to improve the quality of this manuscript.

At the following, the points mentioned by the reviewers will be discussed:

1. The term climate factor should be replaced with ‘meteorological factor’.

[Reply] Thank you for your advice. After careful consideration, we also think it is more reasonable to change " climate factor " to " meteorological factor ". We have made changes in main text (line1, 23, 26, 82, 237, 301). 

2. The author applies GAM to study the relationship for vector density, and long short term memory model (LSTM) for JE incidence with met. Parameters. No justification for adopting two different modeling aspect.

[Reply] Thank you for your advice. This study analyzed the correlation between meteorological variables and the incidence of JE by using the ZIGAM. With the help of the research results, we use LSTM to predict the occurrence of JE in the next year, so that the prediction will be more accurate. We added a comparison of the LSTM and GAM models in S3 table and Figure 5.

3. The predictive ability of the model should be written abstract line, and author avoid to do this. 

[Reply] Thank you for your advice. We have made changes in the figure of the predictive ability of the model (Figure 5).

4. The Introduction section is not structured. Globally published literature is missing. The hierarchy should be, global aspect, then continental, then country, then province/district. The author fails to perform this hierarchy. 

[Reply] Thank you for your advice. We have made changes according to your advice (line 37-49). 

5. Authors wrote, “However, no studies on the prediction of the incidence of JE, which is based on 84 factors such as mosquito density and climate, have been carried out” (pg. 4. line (83-84). In what context the author states this sentence, global or local. Around the globe, researchers have already contributed the stochastic modeling skill for association of JE cases and meteorological parameters (Lin et al., 2012; Pisudde et al., 2017). In what respect the author would like to state that. Justify and clarify. 

[Reply] Thank you for your advice. We have made corresponding amendments according to your advice (line 70-79).

6. Line #83-88. Is there only the meteorological factors that affect JE vector density for JE incidences? The several other factors like land use land cover changes, built dam, water bodies, socioeconomic status of the exposed population, intervention schemes/policy. Justify, why authors do not take into account, all these? 

[Reply] Thank you for your advice. Due to the problem of data acquisition, our research did not consider these factors that may have a certain impact on the occurrence of JE. We also mentioned these factors in the discussion section, requiring further study and improvement in the future work (line 295-299).

7. Why author opt for stochastic approach over the deterministic approach, the compartment modeling (deterministic) is also another way to model perfectly. Justification needed.

[Reply] Thank you for your advice. We observed that the compartment model was used for some deterministic problems and required idealized hypothetical conditions. Therefore, the existing compartment model has certain defects in predicting the trend of infectious disease (line 75-79).

8. Line #89, instead of Methods, there should be Study Area, Data and Methods. The further subsection may be there. 

[Reply] Thank you for your advice. We have made corresponding amendments according to your advice (line 86).

9. The temporal resolution of data is missing in the ‘Data Collection’ section. For the ease of the scientific community, this should be there. A table may be presented here, for data sources of JE incidences as well as meteorological variables, along with the temporal resolution (daily/weekly/fortnightly/monthly/yearly). 

[Reply] Thank you for your advice. We have added the corresponding content according to your advice (line 113-117).

10. The models’ outcome in the result section is not expressed much. The author was only direct to see the tables and figures. The crucial observations are not expressed. 

[Reply] Thank you for your advice. We have added the corresponding content according to your advice (line 195-204 and line 229-233).

11.The result and methods are very inconsistent. The result outcomes are self-contradictory. The result need to restructure. 

[Reply] Thank you for your advice. We have modified the content of the results (line 195-204 and line 229-233).

12. Discussion is poorly written. Author try to make this very lengthy instead of technical discussion and suggestion. The scientific approach is missing in the discussion part.

[Reply] Thank you for your advice. We have made major revisions to the discussion (line 235-299).

13. The conclusion section is missing. 

[Reply] Thank you for your advice. We have added the conclusions (line 301-310).

Response to Reviewer #2:

We would like to thank the reviewer for careful and thorough reading of this manuscript and for the thoughtful comments and constructive suggestions, which help to improve the quality of this manuscript.

At the following, the points mentioned by the reviewers will be discussed:

1. The manuscript presents the results of a correlative analysis between selected climate variables and case incidence of Japanese Encephalitis in central China. The paper addresses a topic that is appropriate for the journal and potentially supplies important information for both basic scientific understanding of environmental influences on disease and for potential application to controlling the disease in this area. The paper compares several methods for deriving correlative models between climate and JE incidence, and does in fact provide the reader with an evaluation of the relative effectiveness of these models. However, the paper is frustratingly vague and difficult to follow at times, particularly in its coverage of the basic methodology and in its summary of how the climate variables are related to disease incidence. 

[Reply] Thank you for your advice. We have introduced the research methods more clearly (line 121-180), and elaborated on the relationship between meteorological factors, mosquitoes and Japanese encephalitis in the discussion section of the manuscript (line 240-272).

2. The major issue with the paper is that it’s just very difficult to follow exactly what the authors did. Figure 2 shows a flow chart of the methods, showing the sources of data and the general flow of the analysis. It would be helpful to include a table in the body of the paper (rather than the supporting materials) that shows exactly w hat meteorological data are included. There are hints in the paper (temperature extrema seem to be one of the variables) but the reader never really knows for sure, and this becomes a problem later in trying to suss out what the findings of the study really are. All of the data are apparently funneled through the ZIGAM method, but the methods section of the paper never really explains what this is or what it does. It is defined on lines 79-80, and some references are cited. Additionally, in lines 130-141 some additional information is provided. Unfortunately, as a person wholly unfamiliar with this technique, this information was of little use. This may not be the case for readers more familiar with this type of analysis, but for those who are not knowledgeable in this area of analysis, the entire methodology description is not very helpful, at best. Since the entire analysis funnels though this technique, I would strongly urge the authors to write a more complete explanation of what this method does, and why it was chosen for this analysis. Otherwise, the reader is left with the frustrating task of searching through several more papers in order to understand what is in this one.

[Reply] Thank you for careful reading of the manuscript. We have made a table in the body of the manuscript, which can accurately display the included meteorological variables (Table 1, line118). Meanwhile, we have further explained the methods of statistical modeling, including ZIGAM , LSTM and other model (line121-180).

3. I had similar issues with the four methods for developing predictive models, LSTM, BPNN, GBM, and SVR. The way the paper is written seems to suggest the LSTM is somehow considered a standard in this type of analysis, and that the other three methods are presented as potentially superior alternatives? Is this correct? IF so, I would suggest that this be made clearer to the reader. On page 7, the paragraph beginning on line 142, some descriptive material for LSTM is provided, but again, in such a way as to be of little help to an unfamiliar reader. I would also note that there is a lot of detail about the other three methods (BPNN, GBM, and SVR) that is simply glossed over. I have only limited experience with neural networks and gradient boosting, but have worked rather extensively with support vector regression, enough to understand that parameter tuning and kernal basis functions are crucial to understanding the effectiveness of them. Absolutely no indication of any of this information is provided. At minimum, one should report the kernal used and the gamma and r parameters. Ideally, the method for determining these parameters should be provided. I suspect that similar information should be provided about the other two methods (GBM, BPNN). Without this type of information, the analysis cannot be replicated, nor does omitting them help other researcher who may wish to use this approach for similar analyses.

[Reply] Thank you for your advice. We have added the detailed explanation of the parameter setting of SVR, BPNN, GBM and GAM (line 167-180).

4. In addition to the lack of relevant detail in the methods section, the actual results also lack some specificity. The purpose of the paper was to model the relationship between climate and JE incidence, and to predict the occurrence of cases. Judging from Figure 5, the latter goal was achieved. Unfortunately, it is not really clear how it was achieved. Which climate variables were most relevant, and how relevant were they? Hard to really tell from the body of the manuscript. In the abstract (lines 27-30) it is noted that mean temperature and humidity are important (as is density of mosquito species in livestock sheds, a variable whose provenance is not very clear). In the body of the manuscript, though, this information is very difficult to separate out. Clearly, the authors have developed a method that is potentially useful for predicting JE in this study area, and potentially useful for researchers who may want to understand this disease (and possibly other, similar, mosquito-borne diseases) in other places. Without much clearer elucidation of their methods, this would be difficult to do. One of my “acid tests” for reading a manuscript is, “given similar data, could I repeat their methods, even if I had never used them before.” In this case the answer is no. Even if I were to track down all of the literature cited to support the methods, It’s still not clear how the authors reached their conclusions.

[Reply] Thank you for your advice. We have supplemented the content in statistical modelling (lines 121-180). In addition, we have agreed to upload a minimal set of anonymous data. The data of this study can be found at https://doi.org/10.17026/dans-xm3-7qdm

---

## [Decision Letter · Decision Letter 1]

11 Jan 2021

PONE-D-20-17377R1

Association between meteorological factors and the prevalence dynamics of Japanese Encephalitis

PLOS ONE

Dear Dr. Tu,

Thank you for submitting your manuscript to PLOS ONE. After careful consideration, we feel that it has merit but does not fully meet PLOS ONE’s publication criteria as it currently stands. Therefore, we invite you to submit a revised version of the manuscript that addresses the points raised during the review process.

Reviewer 1 has raised some minor questions that I think can be addressed very easily.

First of Reviewer 1's comment suggests that you cite a couple of articles. Please know (and it is my recommendation) that you** DO NOT **have to cite the two articles as these papers do not directly inform your work. As far the other comments, please prepare a response and make any changes to the manuscript if you collectively deem necessary.

We appreciate your patience with this review process.

We look forward to receiving your revised manuscript.

Kind regards,

Ram K. Raghavan

Academic Editor

PLOS ONE

Reviewers' comments:

Reviewer's Responses to Questions

**Comments to the Author**

1. If the authors have adequately addressed your comments raised in a previous round of review and you feel that this manuscript is now acceptable for publication, you may indicate that here to bypass the “Comments to the Author” section, enter your conflict of interest statement in the “Confidential to Editor” section, and submit your "Accept" recommendation.

Reviewer #1: All comments have been addressed

Reviewer #2: All comments have been addressed

2. Is the manuscript technically sound, and do the data support the conclusions?

Reviewer #1: Partly

Reviewer #2: Yes

3. Has the statistical analysis been performed appropriately and rigorously? 

Reviewer #1: N/A

Reviewer #2: Yes

4. Have the authors made all data underlying the findings in their manuscript fully available?

Reviewer #1: No

Reviewer #2: Yes

5. Is the manuscript presented in an intelligible fashion and written in standard English?

Reviewer #1: Yes

Reviewer #2: Yes

6. Review Comments to the Author

Reviewer #1: 1. The literature survey still missing the several existing research that have implemented statistical modeling for the association with other Vector Borne Disease like malaria. Author may thoroughly revisit the introduction section and should add some more recent research that focus on Biometeorology. I am suggesting some of the recent findings.

a. Kumar, P., Vatsa, R., Sarthi, P. P., Kumar, M., & Gangare, V. (2020). Modeling an association between malaria cases and climate variables for Keonjhar district of Odisha, India: a Bayesian approach. Journal of Parasitic Diseases. https://doi.org/10.1007/s12639-020-01210-y

b. Kumar, P., Pisudde, P. M., Sarthi, P. P., Sharma, M. P., & Keshri, V. R. (2017). Acute encephalitis syndrome and Japanese Encephalitis, status and trends in Bihar State, India. THE NATIONAL MEDICAL JOURNAL OF INDIA, 30, 317–320. https://doi.org/10.4103/0970-258X.239070

2. How do the meteorological variable play role in the life cycle of JE virus Vector, author should must address here.

3. Do different geography have different threshold for the rainfall, temperature, relative humidity, etc. Author should must discuss in the discussion section.

4. How the build model may be operationalized for public services? Is there any such scope? Brief in short, the application of your model in real time.

5. Author say there is a positive effect on the outbreak with a lag 1-month of some meteorological variable. Is this really feasible, as the mosquito complete the life cycle within a month. A justification should be added.

I would like to see the further changes done by the author.

Reviewer #2: My previous concerns with the paper were that it lacked important details on the data used and the analysis methods. These have now been addressed. There is no further revision needed.

7. PLOS authors have the option to publish the peer review history of their article (what does this mean?). If published, this will include your full peer review and any attached files.

Reviewer #1: **Yes: **Praveen Kumar

Reviewer #2: No

---

## [Author Response · Author response to Decision Letter 1]

14 Feb 2021

February 9, 2021

PLoS ONE

Dear Editors,

On behalf of my co-authors, we thank you very much for giving us an opportunity to revise our manuscript, we appreciate the editor for their meticulous working attitude.

We have replied the point-by-point to the reviewers' comments and revised the manuscript carefully. the comments of all reviewers are all valuable and very helpful for revising and improving our paper. 

We hope that the revised manuscript could meet the high standard of your prestigious journal.

Best regards,

Taotian Tu (taotiantu@sina.cn)

Chongqing Municipal Center for Disease Control and Prevention, Chongqing, China;

Wenge Tang (twg@cqcdc.org)

Chongqing Municipal Center for Disease Control and Prevention, Chongqing, China.

Response to Reviewer #1:

We would like to thank the reviewer for careful and thorough reading of this manuscript and for the thoughtful comments and constructive suggestions, which help to improve the quality of this manuscript.

At the following, the points mentioned by the reviewers will be discussed:

1.The literature survey still missing the several existing research that have implemented statistical modeling for the association with other Vector Borne Disease like malaria. Author may thoroughly revisit the introduction section and should add some more recent research that focus on Biometeorology. I am suggesting some of the recent findings.

[Reply] Thank you for your advice. We have added most recent publications on Biometeorology in the introduction (line 44, 70, line 345-347 and line 369-371). 

2. How do the meteorological variable play role in the life cycle of JE virus Vector, author should must address here.

[Reply] Thank you for your advice. We have added the corresponding content according to your advice (line 252-253, line 276-286).

3. Do different geography have different threshold for the rainfall, temperature, relative humidity, etc. Author should must discuss in the discussion section.

[Reply] Thank you for your advice. We have revised the corresponding content of the discussion (line 289-296).

4. How the build model may be operationalized for public services? Is there any such scope? Brief in short, the application of your model in real time.

[Reply] Thank you for your advice. We have made changes according to your advice (line 299-301, line 330-332). 

5.Author say there is a positive effect on the outbreak with a lag 1-month of some meteorological variable. Is this really feasible, as the mosquito complete the life cycle within a month. A justification should be added.

[Reply] Thank you for your advice. We have added the corresponding content according to your advice (line 276-286).

---

## [Editor Report · Decision Letter 2]

18 Feb 2021

Association between meteorological factors and the prevalence dynamics of Japanese Encephalitis

PONE-D-20-17377R2

Dear Dr. Tu,

We’re pleased to inform you that your manuscript has been judged scientifically suitable for publication and will be formally accepted for publication once it meets all outstanding technical requirements.

Kind regards,

Ram K. Raghavan

Academic Editor

PLOS ONE
---

## [Editor Report · Acceptance letter]

22 Feb 2021

PONE-D-20-17377R2 

Association between meteorological factors and the prevalence dynamics of Japanese Encephalitis 

Dear Dr. Tu:

I'm pleased to inform you that your manuscript has been deemed suitable for publication in PLOS ONE. Congratulations! Your manuscript is now with our production department. 

Kind regards, 

on behalf of

Dr. Ram K. Raghavan 

Academic Editor

PLOS ONE